# Lattice Distortion in $Co_3O_4$/$Mn_3O_4$-Guided Synthesis via Carbon Nanotubes for Efficient Lean Methane Combustion

Xinfang Wei [1,*], Ke Yang [1], Qinghan Zhu [1], Jinlong Li [1], Jian Qi [2,3,*] and Haiwang Wang [1,*]

1 Key Laboratory of Dielectric and Electrolyte Functional Material Hebei Province, Northeastern University at Qinhuangdao, Qinhuangdao 066004, China; y1210620501@163.com (K.Y.); zhuqinghan0110@163.com (Q.Z.); ljl218126@126.com (J.L.)
2 State Key Laboratory of Biochemical Engineering, Institute of Process Engineering, Chinese Academy of Sciences, Beijing 100190, China
3 School of Chemical Engineering, University of Chinese Academy of Sciences, Beijing 100049, China
* Correspondence: weixinfang@neuq.edu.cn (X.W.); jqi@ipe.ac.cn (J.Q.); whwdbdx@neuq.edu.cn (H.W.)

**Abstract:** In this paper, with the synergistic effect of C, Co and O elements, and with a one-dimensional carbon nanotube (CNT) as the structure guide agent, a two-dimensional $Co_3O_4$ nano-sheet with high catalytic activity was prepared, and its catalytic activity was further improved by adding a manganese element. By controlling the annealing time, a two-dimensional $Co_3O_4$-based nano-sheet with a regular arrangement of atoms on the surface was gradually formed during the oxidation process of CNT. Then, the lattice distortion of $Co_3O_4$ was caused by doping manganese. The interaction between Mn and Co promotes the cycle capacity of Mn/Co redox pair and the generation of reactive oxygen species, which is conducive to an improvement in catalytic activity. Finally, under different catalytic conditions, the 2D $Co_3O_4$-based catalysts all showed stable catalytic performances, among which methane flow rate had a great influence. When the airspeed was controlled within the range of 42,000 mL·g$^{-1}$·h$^{-1}$, the methane conversion rate could still reach 90% (450 °C).

**Keywords:** lattice distortion; $Co_3O_4$/$Mn_3O_4$; carbon nanotube; structure guide agent; lean methane combustion

## 1. Introduction

In coal mining, gas turbines, and other industrial applications, there are many methane emissions, and the traditional high-temperature incineration method leads to secondary pollution, so the realization of efficient methane conversion has become a problem to be solved [1]. With the increasing demand for sustainable energy and green energy, the use of catalysts to catalyze methane is an efficient and effective way to purify waste gas. It is well known that spinel-phase $Co_3O_4$ can catalyze the deep oxidation of low concentration methane, which is considered as one of the most promising transition oxide catalysts [2–4]. Studies have shown that a $Co_3O_4$ catalyst can give full play to the size effect, the nanometer size of which can further improve the catalytic activity [5–7]. Moreover, even with nano-crystalline catalysts of the same size, the catalytic efficiency of methane is still greatly different due to their different morphologies [8–12]. This is mainly due to the different arrangement of atoms on the surface of catalysts with different morphologies, which leads to great differences in the number of dangling bonds on the crystal surface and the content of vacancy oxygen on the surface and, finally, affects the catalytic activity [13]. Among the various common nano-topologies, the two-dimensional oriented $Co_3O_4$ nano-catalyst has unique catalytic properties due to its large specific surface area, regular atomic arrangement on the surface and large number of dangling bonds [14,15].

Therefore, the design and preparation of $Co_3O_4$ nanomaterials with special morphology, and their alloys, have attracted wide attention [16]. Yang et al. developed the aqueous phase extraction method to prepare nanometer quantum dots and their alloys [17], which

have very regular microstructures and excellent catalytic properties. Hu et al. prepared a two-dimensional flaky $Co_3O_4$ catalyst with an exposed {112} crystal surface using the hydrothermal method and realized the complete catalysis of low-concentration methane (2% $CH_4$ + 98% air) under the temperature of 375 °C [13]. Also, they prepared two-dimensional $Co_3O_4$ nano-sheets with high species content of surface oxygen using the hydrothermal method [15], which can completely catalyze methane gas (1.6% $CH_4$ + 7.0% $O_2$ + 91.4% $N_2$) at 473 °C. Chen et al. also successfully prepared a two-dimensional hexagon $Co_3O_4$ catalyst with an exposed {111} crystal surface using the hydrothermal method and achieved 100% methane conversion (0.2% $CH_4$ + 99.8% air) at 500 °C [10].

In recent years, studies have shown that, under the synergistic effect of elements, not only can new materials with unique structure be prepared, but novel materials with excellent performance can also be prepared [18–20]. Research has shown that the composite material obtained by combining transition metal oxides with $Co_3O_4$ can combine the advantages of different substances. It has better electrochemical performance when used as an anode of a lithium battery [21–23]. In 2010, Song et al. found that metal sheets loaded inside the CNT could diffuse to the position of carbon atoms inside the CNT under thermal oxidation, thus forming metal oxide nanotubes [24]. The metal/metal oxide CNT nanotubes and mesoporous metal oxide nanotubes are expected to find many applications, such as in lithium ion batteries, catalysis, magnetic drug delivery and gas sensing [25]. In 2017, Bol et al. successfully prepared single-walled CNT, using $Co_3O_4$ film as the substrate, through reduction, vapor deposition and other processes [26]. In this process, Co acts as a catalyst, and the CNT prepared by this process have the characteristics of good regularity and strong controllability [27,28].

In this work, the 2D $Co_3O_4$ nano-sheet catalyst, and its $Co_3O_4/Mn_3O_4$ composite oxide with lattice distortion were prepared by using 1D CNT as the structure guide and under the coordination of C, O and Co elements. This kind of material had a simple preparation process, novel microstructure and an excellent catalytic oxidation performance of methane, which was tested on a fixed-bed evaluation device (Figure S1) and provided a new method for the design and preparation of new $Co_3O_4$-based nanomaterials [29].

## 2. Results

### 2.1. Structural Properties of $Co_3O_4$ Nano-Sheet

Figure 1 is the transmission electron microscope (TEM) image of CNT and the two-dimensional $Co_3O_4$ nano-sheet (the preparation method is shown in Section 4.1). In Figure 1a, 0.34 nm is the corresponding (002) crystal surface. In Figure 1b, the diameter of CNT is about 20 nm. In Figure 1c, 0.46 nm and 0.24 nm, respectively, corresponded to the (111) and (311) crystal surfaces of $Co_3O_4$, mainly based on the (111) crystal plane, which means that the atomic arrangement of the exposed crystal plane of $Co_3O_4$ is relatively regular. According to Figure 1d, the diameter and size of the $Co_3O_4$ nano-sheet can be further determined to be about 20 nm, close to the diameter of CNT [30].

The TG diagram of CNT, $Co(NO_3)_2 \cdot 6H_2O$ and $Co_3O_4$ nanometer sheet precursors in the air atmosphere are shown in Figure 2a. Among them, the CNT begins to be oxidized at about 500 °C and is completely oxidized at about 650 °C. $Co(NO_3)_2 \cdot 6H_2O$ began to be oxidized at about 100 °C, and is completely oxidized at about 300 °C. The TG curve of the precursor of $Co_3O_4$ nano-sheet indicates that, as cobalt nitrate is decomposed and part of the cobalt is oxidized to $Co_3O_4$, the ratio of the $Co^{3+}$ and $Co^{2+}$ content in the sample is 0.79, the oxidation temperature of CNT decreased and it can be completely oxidized at about 500 °C; the existence of $Co_3O_4$ prompted the complete oxidation of CNT nearly 200 °C in advance. The pore size distribution diagram of CNT and the $Co_3O_4$ nano-sheet (Figure 2b) indicates that the sample of the $Co_3O_4$ nano-sheet has a wide range of pore sizes, which is significantly different from the pore volume of CNT and has a large difference in specific surface area. The BET characterization data of the two samples are given in Table 1.

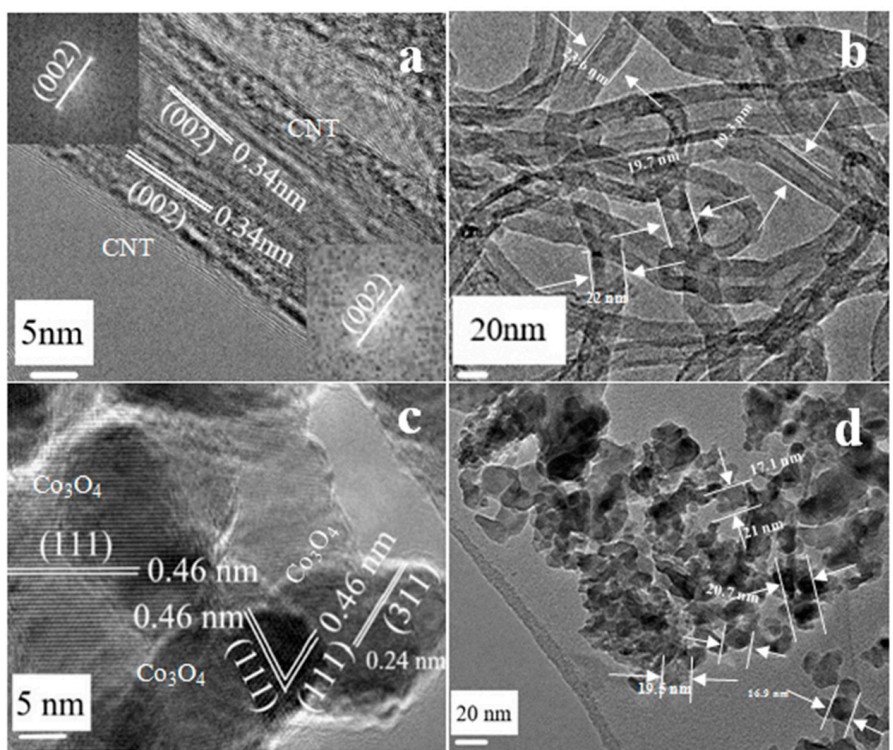

**Figure 1.** (**a**) HRTEM images of CNT; (**b**) TEM images of CNT; (**c**) HRTEM images of $Co_3O_4$ nanosheet; (**d**) TEM images of $Co_3O_4$ nano-sheet.

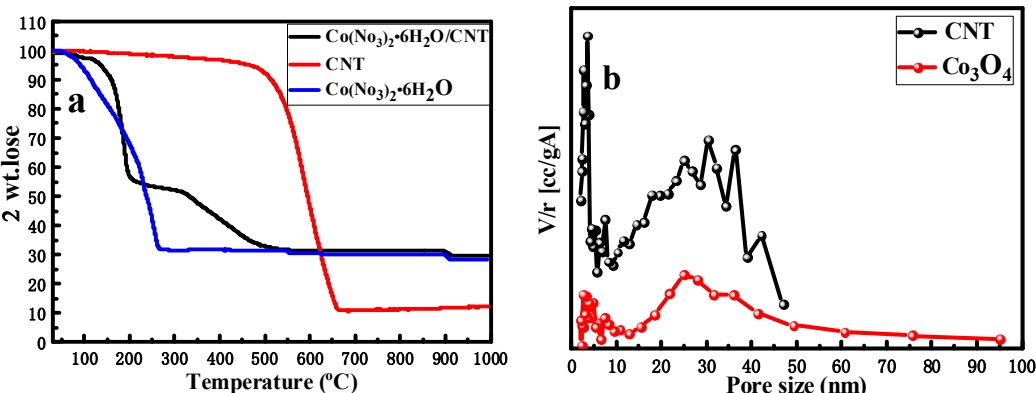

**Figure 2.** (**a**) TG patterns of $Co(NO_3)_2 \cdot 6H_2O$, CNT and $Co(NO_3)_2 \cdot 6H_2O$/CNT; (**b**) pore size distributions of $Co_3O_4$ nano-sheet and CNT.

**Table 1.** Pore parameters of different catalysts.

| Sample | Average Pore Size (nm) | Surface Area ($m^2 g^{-1}$) | Total Pore Volume ($cm^3 g^{-1}$) |
|---|---|---|---|
| CNT | 16.12 | 128.51 | 0.52 |
| $Co_3O_4$ | 34.16 | 28.50 | 0.24 |
| Co0.9/Mn0.5 | 24.70 | 64.81 | 0.44 |
| Mn0.5/CNT | 25.28 | 144.34 | 0.77 |

### 2.2. Analysis of the Formation Process of $Co_3O_4$ Nano-Sheet

It has been observed that $Co_3O_4$ has a significant effect on the stability of CNT through the characterization analysis in Figure 2. The influence of $Co_3O_4$ on the thermochemical stability of CNT is also an important reason for the formation of two-dimensional $Co_3O_4$

nano-sheets, so the influence of $Co_3O_4$ on the oxidation behavior of CNT was analyzed in detail [31].

To help further understand the effect of $Co_3O_4$ on the oxidation of CNTs, we annealed the CNT samples that were loaded with cobalt nitrate at 400 °C for different times (1 min, 3 min, 6 min, 10 min, 30 min and 60 min), and four samples (1 min, 6 min, 30 min and 60 min) were selected for detailed characterization analysis. Figure 3a–d are SEM diagrams of the four samples. It can be found that the structure of CNT changes significantly with the passing of time, that is, CNT changes from the initial tubular structure to the spherical structure assembled by the two-dimensional nano-sheet. The change in structure indicates that the oxidation degree of CNT increases gradually. SEM diagrams of other samples are shown in Figure S2.

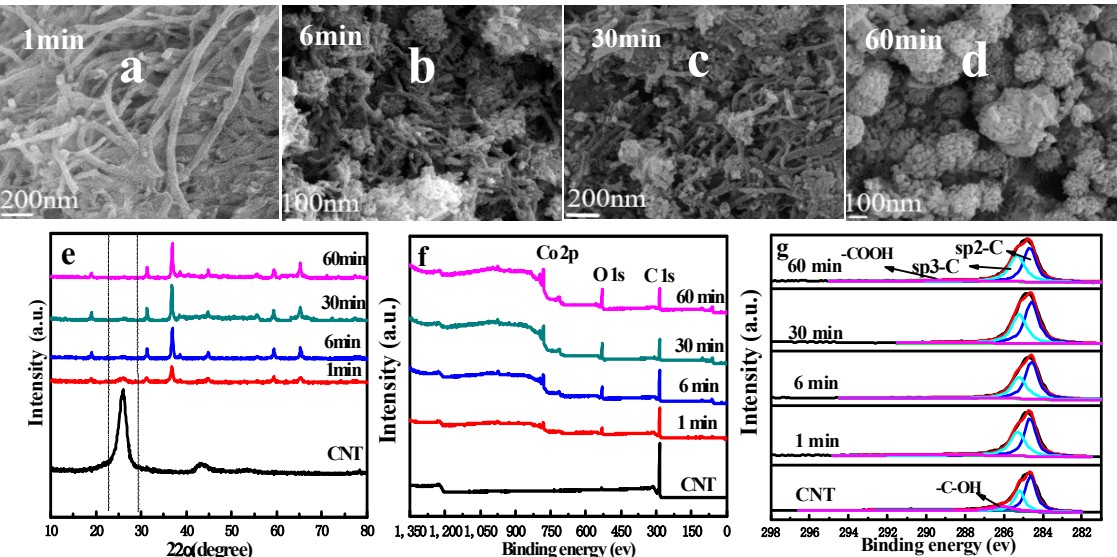

**Figure 3.** (**a**–**d**) SEM images of $Co_3O_4$ nano-sheet prepared at different holding times; (**e**) XRD patterns of $Co_3O_4$ nano-sheet prepared at different annealing times; (**f**) X-ray photoelectron spectrum of $Co_3O_4$ nano-sheet prepared at different annealing times; (**g**) peak deconvolutions of C 1s core lines in XPS spectra of $Co_3O_4$ nano-sheet prepared at different annealing times.

The XRD characterization of the four samples is shown in Figure 3e. The diffraction peaks of $Co_3O_4$ nano-sheet samples well fit the data of the plane diffraction standard of the cubic spinel $Co_3O_4$ (111), (220), (311), (222), (400), (422), (511) and (440), and no CNT diffraction peaks were observed. The diffraction peak of CNT gradually weakened, and the diffraction peak of the sample was completely consistent with that of the structure of the spinel $Co_3O_4$ (JCPDS NO: 76-1802) after 60 min of air burning, further indicating the transition oxidation of CNT. The XRD diagrams of the remaining samples are shown in Figure S3.

Figure 3f shows the full spectrum of XPS of the four samples. Even after 60 min of air burning, the presence of higher contents of carbon elements can still be found, which is attributed to the existence of more stable carbon impurities in the oxidation process. The atomic mass percentages of the three elements, Co, O and C, are given in Table 2, where the content of C gradually decreases with the extension of the blanking time, and the elements Co and O show an opposite trend, which is consistent with the trend of SEM and XPS full spectrum. The carbon bonds of carbon elements in different samples were mainly $sp^2$-hybridization bonds and $sp^3$-hybridization bonds, and the carbon peak of the initial CNT was mainly $sp^2$-hybridization. In Figure 3g, the $sp^3$-hybridization degree gradually increases with the extension of the air burning time, indicating that the stability of CNT gradually deteriorates, and the degree of oxidation gradually deepens [32]. However, two-dimensional $Co_3O_4$ nano-sheets are gradually formed along with the change in CNT

structure, which proves that sufficient air burning time is an important prerequisite for the formation of $Co_3O_4$ nano-sheets.

**Table 2.** XPS Results of CNT and $Co_3O_4$ nano-sheet catalysts prepared at different holding times (surface atomic composition (%), peak position (eV).

| Sample | Surface Atomic Composition (%) | | | Peak Position (eV) |
|---|---|---|---|---|
| | **Co** | **O** | **C** | **C 1s** |
| 60 min | 9.62 | 26.31 | 64.07 | 284.8 |
| 30 min | 7.31 | 23.56 | 69.12 | 284.8 |
| 6 min | 4.82 | 15.04 | 80.14 | 284.6 |
| 1 min | 2.99 | 10.33 | 86.68 | 284.8 |
| CNT | - | 2.31 | 97.69 | 284.8 |

*2.3. Structural Properties of Mn0.5/CNT and Co0.9/Mn0.5 Nano-Sheets*

Recent studies have shown that Co-Mn composite oxide catalysts have higher catalytic activity [33]. Li et al. reported that Co5Mn1 catalysts prepared by co-precipitation processes have higher catalytic activity on methane combustion than pure $Co_3O_4$ [34]. Tian et al. reported that there was a good synergistic effect between $Co_3O_4$ and $MnO_2$, which promoted the formation of reactive oxygen species by changing the lattice structure of $Co_3O_4$ [35]. Marco et al. also reported the following reactivity trends observed in total oxidation of VOCS: $Mn_3O_4 > Mn_2O_3 > Mn_xO_y$ [36]. This indicates that the synergistic effect between $Co_3O_4$ and oxygen species with different valence manganese can not only optimize the structure of the catalyst, but also improve the catalytic performance. However, the preparation of a 2D nano-structure co-manganese composite catalyst for methane oxidation has not been reported.

Here, on the basis of the preparation process of $Co_3O_4$ nano-sheet catalysts, different proportions of manganese oxides were further added, and the optimal amount of manganese oxides was determined by preparing $MnO_2$/CNT with different manganese loads. In Figure S4a, the catalytic activities of Mn0.7/CNT and Mn0.5/CNT were not significantly different, so the concentration of manganese nitrate was determined to be 0.5 M.

Figure 4 shows the TEM images of Mn0.5/CNT and Co0.9/Mn0.5 samples. In the HRTEM diagram of the Mn0.5/CNT catalyst (Figure 4a), the crystal plane spacing of 0.34 nm corresponds to the (002) crystal plane of CNT and the crystal plane spacing of 0.31 nm corresponds to the (110) crystal plane of $MnO_2$. In Figure 4b, the tubular structure of CNT was retained in the sample of Mn0.5/CNT, and the sheet size of $MnO_2$ was within the range of 25~40 nm around the CNT in the form of nanosheets. Figure 4c,d show HRTEM and TEM diagrams of Co0.9/Mn0.5. According to Figure 4c, compared with the $Co_3O_4$/CNT catalyst, the crystal lattice fringes of the Co0.9/Mn0.5 sample catalyst were fuzzy, with poor crystallinity and increased crystal plane spacing. Among them, the crystal plane spacing corresponding to $Co_3O_4$ (111) increased to 0.47 nm, indicating that the intervention of manganese ions caused lattice distortion of $Co_3O_4$ [37]. In Figure 4d, the morphology of Co0.9/Mn0.5 sample and $Co_3O_4$ nano-sheet is consistent, and the morphology is still a two-dimensional nano-sheet. The diameter and size of nano-sheets are about 20 nm, but the dispersion is better.

As above, we continued to collect characterization data for the SEM, XRD, XPS, TG and pore size distribution of Mn0.5/CNT and Co0.9/Mn0.5 samples. The SEM and EDS of the $MnO_2$/CNT and $Co_3O_4$/$Mn_3O_4$ samples are given in Figures S5 and S6. It can be seen in Figure S5 that the tubular morphology of the $MnO_2$/CNT catalyst with different manganese loads changed little, while the morphology of the $Co_3O_4$/$Mn_3O_4$ catalyst with different cobalt manganese ratios changed significantly. When Mn: Co = 5:1, the tubular shape of CNT begins to change. When Mn: Co = 1:1, the tubular structure completely

disappears, showing irregular sheets. When Mn: Co = 5:9, the appearance is as shown in Figure 4d.

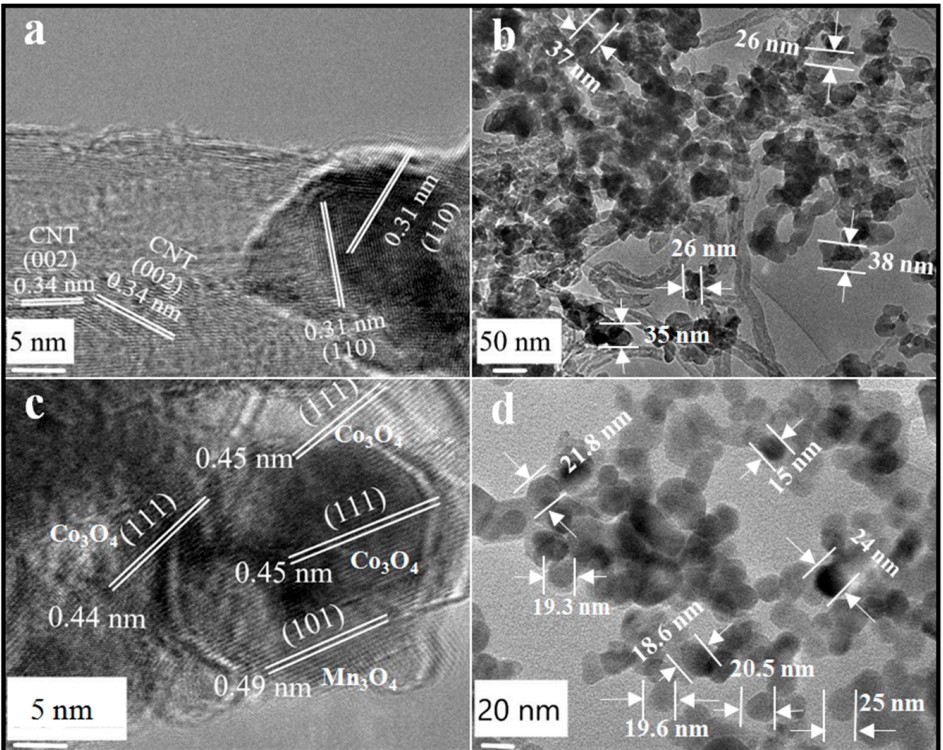

**Figure 4.** (**a**) HRTEM images of Mn0.5/CNT; (**b**) TEM images of Mn0.5/CNT; (**c**) HRTEM images of Co0.9/Mn0.5 nano-sheet; (**d**) TEM images of Co0.9/Mn0.5 nano-sheet.

In Figure S6, the content distribution of manganese and cobalt is positively correlated with the concentration of manganese salt and cobalt salt, which indicates that, before annealing, the adsorption amount of CNT on manganese salt and cobalt salt is proportional to the concentration of salt solution, and the load capacity is good.

In Figure 5a, the diffraction peaks of Mn0.5/CNT samples were shown as the composite peaks of CNT and $MnO_2$, while the diffraction peaks of Co0.9/Mn0.5 samples were highly consistent with those of the $Co_3O_4$ nano-sheet, and no peaks of CNT and $Mn_3O_4$ were observed. According to Figure S7a, the diffraction peaks of $MnO_2$/CNT catalysts with different manganese concentrations are consistent, and there is a change in peak strength which depends on the agglomeration degree of $MnO_2$. In Figure S7b, when the content of $Co_3O_4$ is low, the peak of $Mn_3O_4$ is obvious, but still has a strong oxidation effect on CNT, so the peak of CNT is not obvious. When $Co_3O_4$ content is high, there is no $Mn_3O_4$ diffraction peak, indicating that $Mn_3O_4$ is highly dispersed or partly exists in the lattice of $Co_3O_4$ in an amorphous form [34,38]. It is worth noting that, with the increase in cobalt salt concentration, the half-peak width of the diffraction peak of $Co_3O_4$ gradually widens, and the diffraction peak slightly shifts to the left, which indicates that part of $Mn^{4+}$ is dispersed in the lattice of $Co_3O_4$. Replacing cobalt ions with manganese ions caused lattice distortion of $Co_3O_4$ and increased crystal plane spacing of $Co_3O_4$ [39], which was consistent with the HRTEM analysis results. From the perspective of crystal structure, the crystal field stability of $Mn^{3+}$ is higher than that of $Co^{3+}$ in octahedrons [34], so $Mn^{3+}$ takes the place of $Co^{3+}$ in octahedrons of spinel structure, where the $Mn^{3+}$ ionic radius (0.066 nm) > the $Co^{3+}$ ionic radius (0.065 nm) [35].

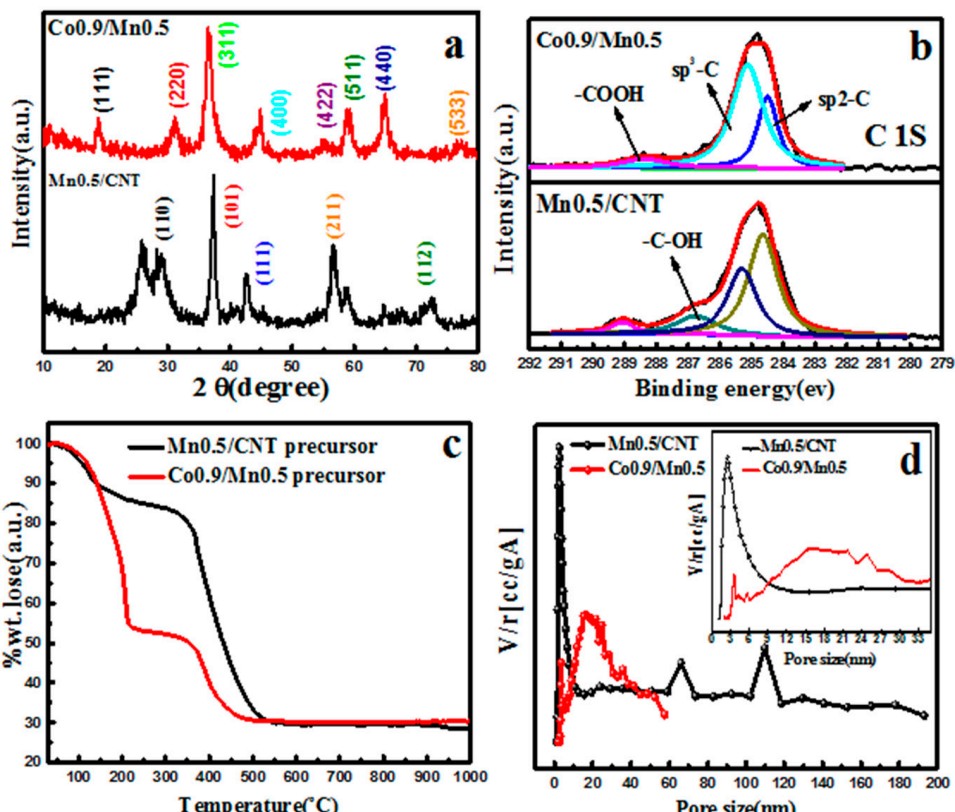

**Figure 5.** (**a**) XRD patterns of Mn0.5/CNT and Co0.9/Mn0.5 nano-sheet; (**b**) XPS spectra of C 1s acquired over Mn0.5/CNT and Co0.9/Mn0.5 nano-sheet; (**c**) TG patterns of Mn0.5/CNT and Co0.9/Mn0.5 nano-sheet precursor; (**d**) pore size distributions of Mn0.5/CNT and Co0.9/Mn0.5 nano-sheet.

Figure 5b shows the XPS spectra of the two samples, and it can be clearly observed that, compared with Mn0.5/CNT, the sp3-hybridized carbon of Co0.9/Mn0.5 sample is more obvious, and there is no -C-OH, indicating that the CNT oxidation in the Co0.9/Mn0.5 sample is more serious. Figure 5c shows the TG diagram of the precursor Mn0.5/CNT and Co0.9/Mn0.5 in the air atmosphere. Among them, the completely weightlessness of CNT and manganese nitrate is at about 550 °C, and the TG curves of Co0.9/Mn0.5 precursor is basically the same with the $Co_3O_4$ precursor. The pore size distribution diagram of $MnO_2$/CNT and $Co_3O_4$/$Mn_3O_4$ (Figure 5d) indicates that $MnO_2$/CNT has a large pore volume and a small pore size range, which may be related to the existence of good tubular CNT. The pore volume of Co0.9/Mn0.5 is small and the pore diameter range is large, but the microscopic pore structure is better than that of the $Co_3O_4$ nano-sheet, so it has a larger specific surface area and pore volume. The pore size distribution and BET characterization data of $MnO_2$/CNT and $Co_3O_4$/$Mn_3O_4$ with different concentration variables are shown in Figure S8 and Table S1.

## *2.4. Activity Evaluation of Catalyst Oxidation of Methane*
### 2.4.1. Catalytic Performance Test of Different Catalysts

So as to evaluate the catalytic activity of the $Co_3O_4$ nano-sheet, Co0.9/Mn0.5 and Mn0.5/CNT catalysts, methane conversion rates were tested at different temperatures ranging from 200 °C to 550 °C. As expected, the $Co_3O_4$ nano-sheet exhibited high catalytic activity, 98% of the methane combustion at 400 °C was implemented and, within the range of 250~350 °C, the methane combustion rate was faster.

Compared with the test results of the $Co_3O_4$ nano-sheet, the Co0.9/Mn0.5 nano-sheet catalyst showed a higher methane combustion conversion rate (Figure 6), and 99% of methane combustion was achieved at 400 °C. The Co0.9/Mn0.5 nano-sheet catalyst began

to catalyze methane combustion (3%) at 200 °C. This confirms that the addition of Mn can significantly improve the catalytic activity [40]. Compared with the $Co_3O_4$ nano-sheet at the same temperature, at 250 °C and 350 °C, the combustion of methane conversion was 10% higher. However, the Mn0.5/CNT catalyst showed lower activity, even though its specific surface area and pore volume were larger.

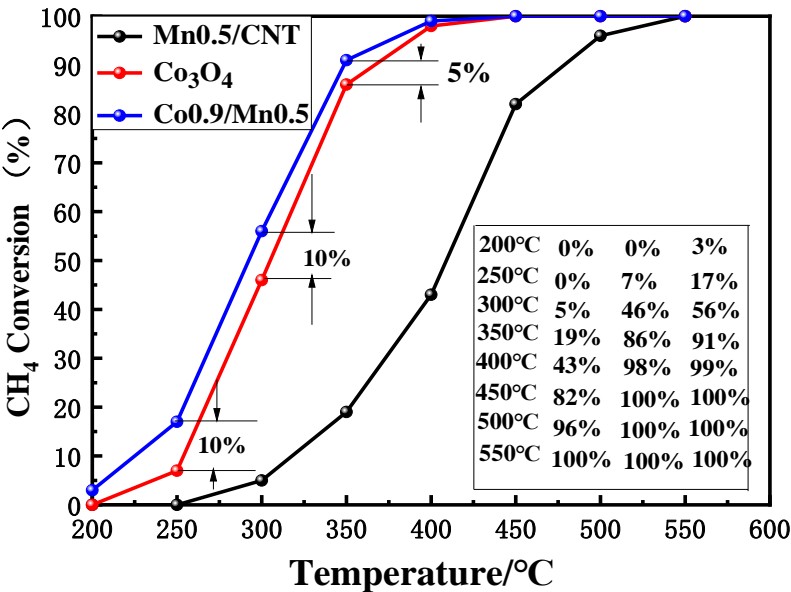

**Figure 6.** $CH_4$ conversion, from 200 to 550 °C, of different catalysts.

Recent studies have shown that higher $Co^{3+}$ and higher adsorbed oxygen content are more favorable for lean methane combustion [33]. In our experiment, compared with the $Co_3O_4$ nano-sheet, doping Mn led to increased values of $Co^{3+}/Co^{2+}$ in the Co0.9/Mn0.5 nano-sheet (confirmed by XPS analysis), which increased the mobility of reactive oxygen species, leading to higher catalytic activity. This fact may be the reason why the catalytic activity of the Co0.9/Mn0.5 nano-sheet was higher than that of the $Co_3O_4$ nano-sheet.

2.4.2. Comprehensive Evaluation of Catalytic Activity of Co0.9/Mn0.5 Catalyst

Next, through simulating the actual situation, the feasibility of the prepared Co0.9/Mn0.5 catalyst in three environments of high temperature conditions (600~900 °C), 450 °C long working time and different methane gas flow rates was validated. Figure 7a shows the catalytic effect of the Co0.9/Mn0.5 nano-sheet catalyst sample at the high temperature condition of 600~900 °C. It was found that the sample did not deactivate at high temperatures, indicating that the sample had the characteristics of high temperature resistance. Next, the stability of the catalyst for a long time at 450 °C was tested. As shown in Figure 7b, the catalytic activity of the catalyst was very stable over 2200 min.

Lastly, the influence of GHSV on catalytic activity was tested in the Co0.9/Mn0.5 nanometer sheet catalyst (Figure 7c). Different methane gas flow rates changed the possibility of contact between methane gas and the catalyst. When the flow rate was too fast, the methane gas may not have reacted completely when leaving the catalyst, which led to a reduction in the removal rate of methane [41]. The fact that methane can still be well burned with a fast flow rate is an important index to evaluate the activity of the catalyst. In order to reduce the influence of error caused by the change in flow velocity, five samples were tested at each flow velocity stage, and the control range of methane flow velocity was GHSV = 6000~60,000 mL/(g h). When the flow rate exceeded GHSV = 12,000 mL/(g h), the catalytic activity decreased. When the flow rate of methane exceeded GHSV = 42,000 mL/(g h), the conversion rate of catalytic methane began to be lower than 90%, indicating that the catalyst had a strong adaptability to the flow rate of methane.

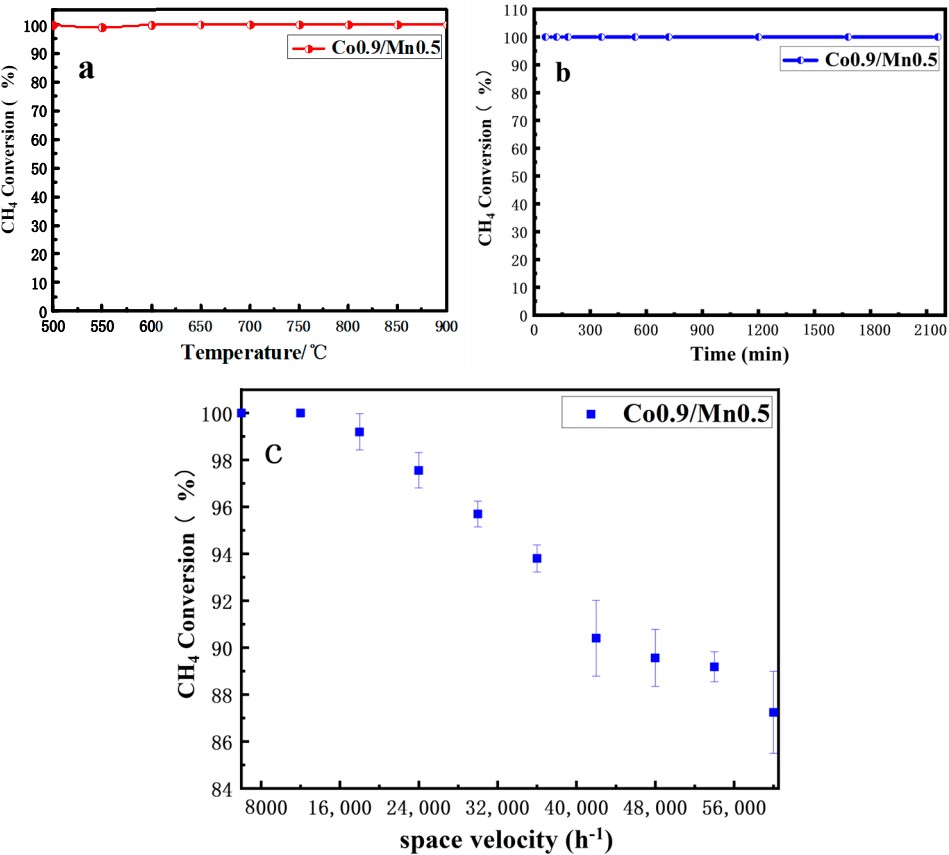

**Figure 7.** (**a**) High temperature resistance to sintering test of $Co_3O_4$ nano−sheet catalyst; (**b**) thermal stability test of $Co_3O_4$ nano−sheet catalyst; (**c**) effect of GHSV on $CH_4$ conversion over $Co_3O_4$ nano−sheet catalyst.

2.4.3. The Study of the Catalytic Combustion Kinetics of Low Concentration Methane Gas

The study of the catalytic combustion kinetics of low concentration methane gas can provide the kinetic basis for the establishment of catalytic methane combustion reactor model and further lay the foundation for the development of the technology of low concentration methane gas catalytic combustion to depth direction. In order to better describe the reaction kinetics and reaction series in the methane catalytic process, the kinetic model was introduced and the following methane catalytic combustion reaction rate equation was obtained through the derivation of the formula. The detailed derivation process is given in the supporting information.

By using the vertical axis as $\ln([M^{-1}s^{-1}])$ and the horizontal axis as $1000/T$, the slope obtained, multiplied by R, gives the activation energy $E_a$. Figure 8 shows the Arrhenius plots of the methane oxidation reaction for the $Co_3O_4$ nano-sheet and the Mn0.5/CNT and Co0.9/Mn0.5 nano-sheet catalysts. Using this formula, within the scope of the 200~325 °C, the activation energy of low concentration methane gas combustion catalyzed by three catalysts was Co0.9/Mn0.5 ($19.53\ kJ\cdot mol^{-1}$) < $Co_3O_4$ ($61.72\ kJ\cdot mol^{-1}$) < Mn0.5/CNT ($80.37\ kJ\cdot mol^{-1}$) from high to low. As can be seen, the activation energies are somewhat lower on the Co0.9/Mn0.5 nano-sheet catalysts, indicating that the addition of $Mn_3O_4$ especially lowered the methane activation barrier [42].

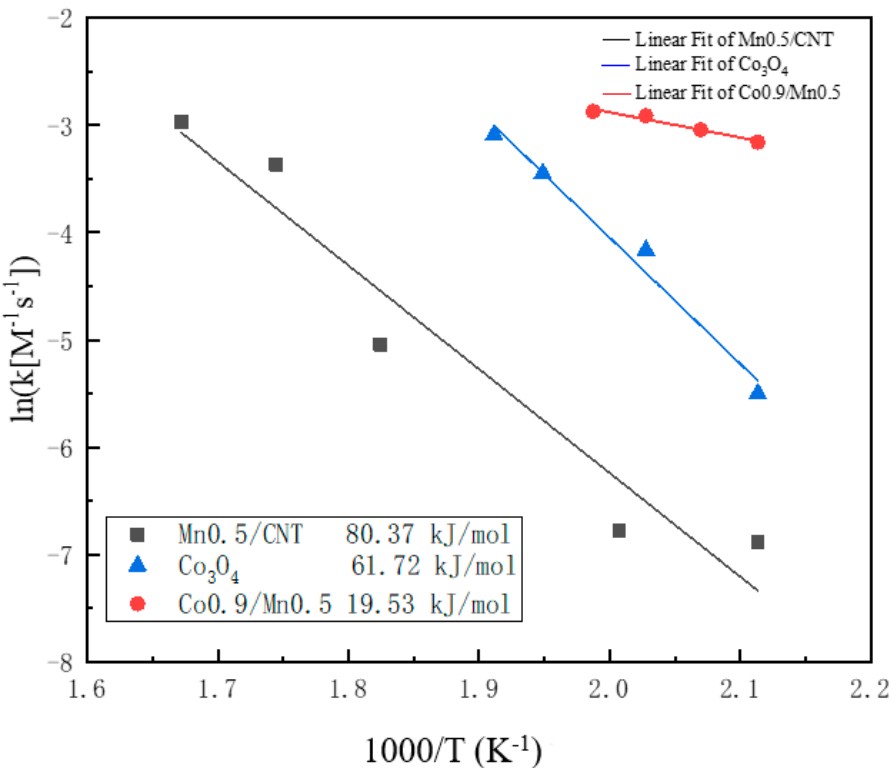

**Figure 8.** Arrhenius-type plots for methane oxidation of $Co_3O_4$ nano−sheet, Mn0.5/CNT and Co0.9/Mn0.5 nano-sheet.

## 3. Discussion

### 3.1. The Interaction of $Co_3O_4$ and $Mn_3O_4$

In order to better understand the influence of $Mn_3O_4$ dopant on the catalytic performance of a $Co_3O_4$ nano-sheet catalyst, XPS analysis was performed. XPS analysis was used to determine the influence of $Mn_3O_4$ doping on the surface chemical state of $Co_3O_4$ before and after doping.

Figure 9 shows the XPS spectra of O 1s, C 1s, Co 2p and Mn $2p_{1/2}$ on the surface of the $Co_3O_4$ nano-sheet, $MnO_2$/CNT and Co0.9/Mn0.5 nano-sheet catalysts. For O 1s (Figure 9a,b), the peak at 530.3 eV corresponds to lattice oxygen ($O_I$), while the peak at 531.6 eV corresponds to adsorbed oxygen ($O_{II}$) [43]. By observing Figure 9a,b, it can be found that the $O_{II}/O_I$ of the Co0.9/Mn0.5 sample was higher than that of the $Co_3O_4$ nano-sheet sample, while the $MnO_2$/CNT sample had more pollutants and adsorbed water. The specific data are shown in Table 3, where $O_I$ and $O_{II}$ were calculated based on the quantitative analysis of the O 1s spectrum, which is attributed to the entry of manganese ions into the lattice of $Co_3O_4$. Because the mobility of $O_{II}$ is stronger than $O_I$, the oxidation ability of $O_{II}$ to methane is stronger than $O_I$, and more $O_{II}$ can generate more oxygen vacancy to improve the catalytic combustion efficiency of methane [44]. The different atomic percentages of the $MnO_2$/CNT and $Co_3O_4/Mn_3O_4$ nano-sheets on the catalyst surface, and their corresponding positions, are given in Table 3. The variation trend of atomic percentages on the catalyst surface was consistent with that of manganese salt and cobalt salt concentration, which is well consistent with Figure S6. The specific value is shown in Table 4. The $O_{II}/O_I$ of the $Co_3O_4/Mn_3O_4$ catalyst were higher than that of $Co_3O_4$, indicating that the intervention of manganese could change the $O_{II}/O_I$ of $Co_3O_4$.

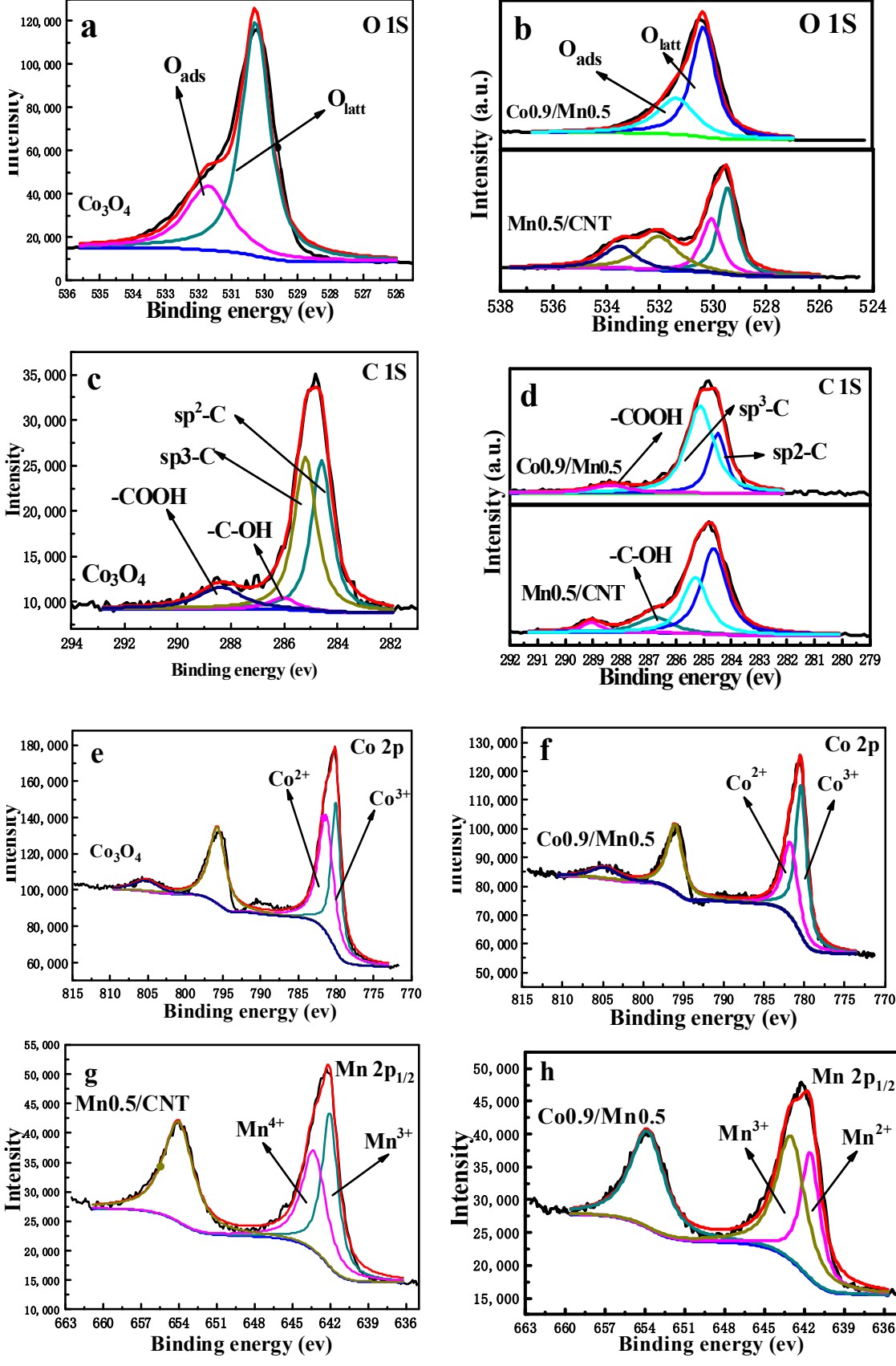

**Figure 9.** XPS spectra acquired of $Co_3O_4$ nano-sheet, Mn0.5/CNT and Co0.9/Mn0.5 nano-sheet: (**a**,**b**) O1s, (**c**,**d**) C1s, (**e**,**f**) Co2p and (**g**,**h**) Mn $2p_{1/2}$.

**Table 3.** XPS results of $MnO_2$/CNT and $Co_3O_4$/$Mn_3O_4$ catalysts (surface atomic composition (%), peak position (eV)).

| Sample | Surface Atomic Composition (%) | | | | Peak Position (eV) | | | |
|---|---|---|---|---|---|---|---|---|
| | **Mn** | **Co** | **O** | **C** | **Mn 2p$_{1/2}$** | **Mn 2p$_{3/2}$** | **Co 2p$_{1/2}$** | **Co 2p$_{3/2}$** |
| Mn0.3/CNT | 8.35 | 0.0 | 22.46 | 69.19 | 653.9 | 642.4 | - | - |
| Mn0.5/CNT | 8.77 | 0.0 | 29.35 | 61.88 | 654.0 | 642.3 | - | - |
| Mn0.7/CNT | 16.58 | 0.0 | 39.8 | 43.62 | 654.3 | 642.6 | - | - |
| Co0.1/Mn0.5 | 13.03 | 2.92 | 34.97 | 49.07 | 653.5 | 642.2 | 796.0 | 780.7 |
| Co0.5/Mn0.5 | 8.98 | 7.68 | 36.8 | 46.53 | 653.9 | 642.0 | 795.8 | 780.5 |
| Co0.9/Mn0.5 | 8.28 | 11.69 | 43.27 | 36.76 | 654.0 | 642.0 | 795.8 | 780.6 |
| $Co_3O_4$ | - | 20.85 | 47.83 | 31.32 | - | - | 795.5 | 780.3 |

**Table 4.** XPS results of $MnO_2$/CNT and $Co_3O_4$/$Mn_3O_4$ catalysts ($O_{II}$/$O_I$, $Co^{3+}$/$Co^{2+}$, $Mn^{4+}$/$Mn^{3+}$, $Mn^{3+}$/$Mn^{2+}$).

| Sample | $O_{II}$/$O_I$ | $Co^{3+}$/$Co^{2+}$ | $Mn^{4+}$/$Mn^{3+}$ | $Mn^{3+}$/$Mn^{2+}$ |
|---|---|---|---|---|
| Mn0.3/CNT | 0.64 | - | 0.96 | - |
| Mn0.5/CNT | 0.69 | - | 0.93 | - |
| Mn0.7/CNT | 0.65 | - | 0.86 | - |
| Co0.1/Mn0.5 | 0.67 | 0.98 | - | 1.18 |
| Co0.5/Mn0.5 | 0.72 | 1.30 | - | 1.27 |
| Co0.9/Mn0.5 | 0.89 | 1.55 | - | 1.46 |
| $Co_3O_4$ | 0.60 | 0.79 | - | - |

The high-resolution C 1s spectra of CNTs are shown in Figure 9c,d, in order to verify the variation of the edge carbon density. As can be seen, four components can be deconvoluted, corresponding to the $sp^2$-hybridized carbon (284.6 eV), $sp^3$-hybridized carbon (285.3 eV), -C-OH (286.1 eV) and -COOH (288.4 eV), respectively [32]. Since the degree of oxidation of CNTs in the three catalysts was different, there was a significant difference in the relative content of $sp^3$-hybridized carbon on the surface of the three catalysts. The relative content of $sp^3$-hybridized carbon on the surface of the Co0.9/Mn0.5 nano-sheet was the highest, indicating that CNTs were most oxidized. Observing the C 1S spectrum of the $MnO_2$/CNT and $Mn_3O_4$/$Co_3O_4$ nano-sheets catalysts in Figure 9c,d, the relative content of $sp^3$-hybridized carbon still had a regular change, that is, the higher the catalytic activity of the catalyst, the more obvious the $sp^3$-hybridization.

In Figure 9e,f, the shape and position of the spectral lines of the $Co_3O_4$ and Co0.9/Mn0.5 nano-sheet catalysts are all typical spinel structures. The peaks at $780.2 \pm 0.2$ and $782.2 \pm 0.2$ eV correspond to $Co^{3+}$ and $Co^{2+}$, respectively, and the small peaks at 789.4 eV belong to the vibration satellite peak of $Co^{2+}$ [45]. In comparison with Figure 9e,f, the ratio of $Co^{3+}$/$Co^{2+}$ in the Co0.9/Mn0.5 nano-sheet sample was higher than that of the undoped $Mn_3O_4$ nano-sheet sample, and the specific data are shown in Table 3. As analyzed by XRD and TEM, $Mn^{3+}$ replaces the $Co^{3+}$ partially located in the octahedron, resulting in lattice expansion. According to the Jahn–Teller effect, the replacement of $Mn^{3+}$ in the octahedron will cause local tetragonal distortion, so the $Mn^{3+}$ occupying the octahedron is not stable and the $Co^{2+}$ near $Mn^{3+}$ will show a trend of transformation to $Co^{3+}$ [33,37]. In the process of conversion from $Co^{2+}$ to $Co^{3+}$, the oxygen vacancy [46] content and oxygen species mobility could be improved, so the catalytic activity of the Co0.9/Mn0.5 catalyst was higher. At the same time, the relative content of $Co^{3+}$/$Co^{2+}$ on the surface of the $Co_3O_4$/$Mn_3O_4$ nano-sheet catalyst with different cobalt-manganese ratios is shown in Figure S9, and the specific data are shown in Table 4. The $Co^{3+}$/$Co^{2+}$ values of the $Co_3O_4$/$Mn_3O_4$ nano-sheet catalysts with different cobalt-manganese ratios were all higher than those of the $Co_3O_4$ nano-sheet catalysts, further indicating that manganese doping can change the relative content of different cobalt valence states.

Figure 9g,h are the Mn $2p_{1/2}$ XPS spectra of Mn0.5/CNT and Co0.9/Mn0.5, respectively. The valence states of Mn in the two samples were different. The $Mn^{4+}$ and $Mn^{3+}$ in the Mn0.5/CNT sample corresponded to 643.4eV and 642eV, respectively (Figure 9g) [45,47], and the $Mn^{2+}$ and $Mn^{3+}$ in the Co0.9/Mn0.5 sample corresponded to 641.6 eV and 643.1 eV, respectively (Figure 9h) [48]. On the surface of the Co0.9/Mn0.5 catalyst, $Mn^{2+}$ and $Mn^{3+}$ could combine with $Co^{2+}$ to form redox pairs of $Co^{2+}$–$O^{2-}$–$Mn^{3+}$, improving the oxygen mobility of the catalyst [35]. In Figure S10, the $Mn^{3+}/Mn^{2+}$ value was also observed to be proportional to the proportion of cobalt and manganese. The $Mn^{3+}/Mn^{2+}$ value of the $Co_3O_4/Mn_3O_4$ catalyst is given in Table 4, and the $Mn^{3+}/Mn^{2+}$ value of Co0.9/Mn0.5 is the largest. According to the catalytic effect, the catalytic activity increases with the increase in $Mn^{3+}$ion content.

### 3.2. Analysis of the Mechanism of $Mn_3O_4$ Promoting the Catalytic Oxidation of Methane by $Co_3O_4$

The characterization analysis shows that the interaction between Mn ions and $Mn_3O_4$ led to lattice distortion in $Co_3O_4$. Specifically, the participation of $Mn^{2+}$ promoted the transformation of $Co^{2+}$ to $Co^{3+}$, but oxygen vacancies needed to be replenished during the process. The lack of oxygen vacancies could be provided by the transformation of $Mn^{3+}$ to $Mn^{2+}$, indicating that $Co^{2+} \rightarrow Co^{3+}$ and $Mn^{3+} \rightarrow Mn^{2+}$ occurred simultaneously. During the transformation of $Co^{2+} \rightarrow Co^{3+}$ or $Co^{3+} \rightarrow Co^{2+}$, a large number of oxygen vacancy defects appeared, effectively adsorbing $O_2$ in methane gas and converting it to lattice oxygen ($O_I$) [1,49]. At the same time, some $O_{II}$ was used for the further oxidation of $Co^{2+}$ and $Mn^{2+}$. This also explains why the relative content of $O_{II}$ in Co0.9/Mn0.5 was higher than that in $Co_3O_4$. Therefore, it can be concluded that the promotion mechanism of Mn ions was mainly achieved by inducing lattice distortion in $Co_3O_4$. In the catalytic oxidation process of methane, the oxygen reduction reaction of $Mn^{3+}/Mn^{2+}$ played a role in buffering the oxygen transport and storage. The oxygen reduction reaction of $Co^{3+}/Co^{2+}$ played a role in transferring oxygen to the catalyst surface and the oxygen vacancies could supplement oxygen molecules in the methane mixture to generate active oxygen. In summary, the promotion mechanism of $Mn_3O_4$ on the catalytic oxidation of methane by $Co_3O_4$ in Figure 10 can be more comprehensively explained and understood by capturing oxygen molecules in the methane mixture to supplement oxygen vacancies and further form active oxygen. The method is as follows [33,35,37]:

$$Co^{2+} + Mn^{3+} \rightarrow Co^{3+} + Mn^{2+} \tag{1}$$

$$Mn^{2+} + O^* \rightarrow Mn^{3+} \tag{2}$$

$$Co^{2+} + O^* \rightarrow Co^{3+} \tag{3}$$

$$O_2 + *_\square \rightarrow O^* \tag{4}$$

Under the interaction of $Mn_3O_4$ and $Co_3O_4$, the formation of reactive oxygen species can be accelerated by enhancing the oxygen mobility on the catalyst surface ($*_\square$ represents catalyst active center, $O^*$ represents oxygen vacancy).

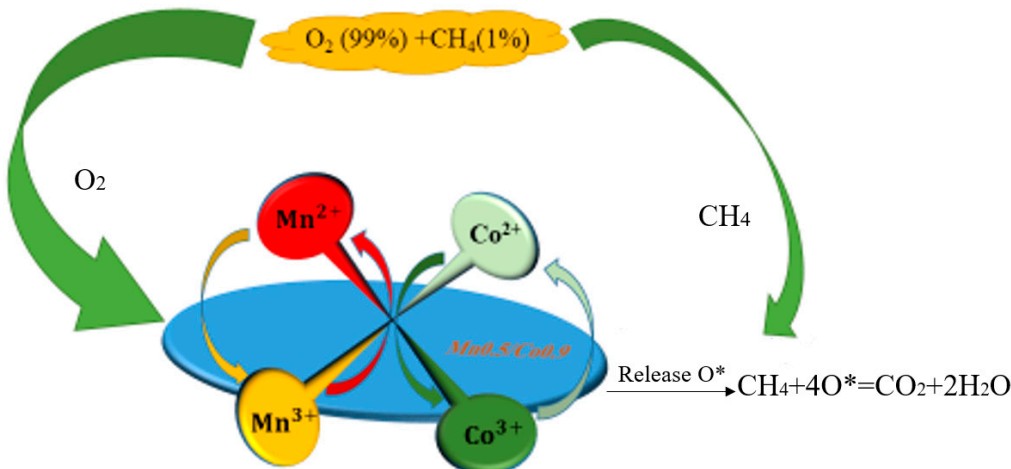

**Figure 10.** The mechanism of $Mn_3O_4$ promoting the catalytic oxidation of $CH_4$ by $Co_3O_4$ (O* represents oxygen vacancy).

## 4. Materials and Methods

### 4.1. Preparation of Catalysts

Preparation of $Co_3O_4$ nano-sheet: $Co_3O_4$ nano-sheets were prepared using the hard template method. An amount of 0.9 M cobalt nitrate solution of 25 mL was prepared. A total of 0.15 g CNT (Manufactured by Shenzhen Guoheng Qihang Technology Co., Ltd., Shenzhen, China) was immersed in $Co(NO_3)_2 \cdot 6H_2O$ solution and boiled at 100 °C for 1 h. The cobalt nitrate solution was then adsorbed and left overnight. Finally, they were blown dried under 80 °C for 6 h, and annealed under 400 °C for 3 h.

Preparation of $MnO_2/CNT$ and $Co_3O_4/Mn_3O_4$: $MnO_2/CNT$ and two-dimensional $Co_3O_4/Mn_3O_4$ nano-sheets were prepared using the hard template method. As is typical, 25 mL composite solution of manganese nitrate solution and manganese nitrate cobalt was prepared. Then, 0.15 g CNT was weighed and soaked in the above solution and boiled for 1 h at 100 °C. The solution was then sucked out and left overnight. Finally, they were blown dried under 80 °C for 6 h, and annealed under 400 °C for 3 h. In the process of preparing $MnO_2/CNT$, the catalysts prepared by manganese nitrate solutions with different concentrations were expressed by MnX/CNT, where X represented manganese nitrate solutions with concentrations of 0.3 M, 0.5 M and 0.7 M. In the preparation of two-dimensional $Co_3O_4/Mn_3O_4$ nano-sheet, CoY/Mn0.5, Mn0.5 represents manganese nitrate solution with a solubility of 0.5 M, and Y represents $Co(NO_3)_2 \cdot 6H_2O$ solution with a concentration of 0.1 M, 0.5 M and 0.9 M.

### 4.2. Catalyst Characterization

Morphological and microstructure analyses of the different types of catalysts were performed using a scanning electron microscope (SEM; ZEISS SUPRA–55, Zeiss, Munich, Germany) at an acceleration voltage of 5 kV. The phase composition was analyzed utilizing a powder X-ray diffraction system with Cu Kα radiation (XRD; DX2500 Dandong Fangyuan, Dandong, China). The specific surface areas (SSA), the pore size distribution (PSD) and pore volume were determined by specific surface area analyzer (SSA; SSA-4000 Biaode, Lanzhou, China). TG was analyzed by thermal gravimetric analyzer (TG; HCT-2 Beijing Hengjiu Scientific Instrument Factory, Beijing, China). Surface chemical state analysis of the different types of catalysts was characterized by X-ray photoelectron spectroscopy (XPS; ESCALAB250 Thermo VG, Waltham, MA, USA). The charging effect was corrected by referencing the binding energy of C 1 s at 284.6 eV.

### 4.3. Catalytic Performance Measurements

The catalytic performance for $CH_4$ combustion was tested in a fixed-bed quartz reactor (i.d.4 mm) packed with 0.1 g catalyst. The reaction temperature was controlled by intelligent

temperature regulator. The low concentration methane gas, containing 0.1 vol.% $CH_4$ in air, was supplied to the catalyst bed through a mass flow controller at a gas hourly space velocity (GHSV) of 6000 mL·g$^{-1}$·h$^{-1}$. The gas compositions were also analyzed by a GC-7890 gas chromatograph equipped with FID and TCD detectors, the model of the activity evaluation device is shown in Figure S1. The conversion of $CH_4$ was calculated using the following formula:

$$X(\%) \ = \ (C_{in} - C_{out})/C_{in} \ \times 100\% \tag{5}$$

where X presents the $CH_4$ conversion, $C_{in}$ stands for the initial $CH_4$ concentration in the inlet and $C_{out}$ stands for the $CH_4$ concentration in the outlet.

## 5. Conclusions

In this work, for the first time, a two-dimensional $Co_3O_4$ nano-sheet catalyst was successfully prepared using CNT as a hard template. The two dimensional Co0.9/Mn0.5 composite catalyst with higher catalytic activity was prepared by doping $Mn_3O_4$. During the preparation of the $Co_3O_4$ nano-sheet, the thermal stability of CNT and the structural changes of $Co_3O_4$ were further understood by controlling the air burning time. The microstructure characterization and surface chemical states of the three types of catalysts proved that the high catalytic activity of the Co0.9/Mn0.5 catalyst was mainly attributed to the following reasons: (1) the interference of $Mn^{3+}$ causes lattice distortion of $Co_3O_4$ and reduces the agglomeration degree of the catalyst. (2) The regular arrangement of atoms on the surface of the exposed crystal face is one of the important reasons for the high catalytic activity. (3) The interaction between Mn and Co promotes the cycle capacity of Mn/Co redox pairs and produces more species of reactive oxygen species. By comparing the activation energy of methane gas combustion catalyzed by three catalysts, it can be found that the Co0.9/Mn0.5 catalyst with high catalytic activity can significantly reduce the activation energy of methane combustion. Meanwhile, the catalytic activity of the Co0.9/Mn0.5 catalyst remained stable under different catalytic conditions.

**Supplementary Materials:** The following supporting information can be downloaded at: https:// www.mdpi.com/article/10.3390/catal13071112/s1, Figure S1: Schematic diagram of the experimental set-up; Figure S2: SEM images of $Co_3O_4$ nano-sheet prepared at annealing times of 3 and 10 min; Figure S3: XRD patterns of $Co_3O_4$ nano-sheet prepared at different annealing times, and JCPDS card is from ref. 50; Figure S4: $CH_4$ conversion from 200 to 550 °C over (A) $MnO_2$/CNT and (B) $Co_3O_4$/$Mn_3O_4$ catalysts; Figure S5: SEM images of $MnO_2$/CNT and $Co_3O_4$/$Mn_3O_4$ catalysts; Figure S6: EDS images of $MnO_2$/CNT and $Co_3O_4$/$Mn_3O_4$ catalysts; Figure S7: XRD patterns of $MnO_2$/CNT and $Co_3O_4$/$Mn_3O_4$ catalysts; Figure S8: Pore size distributions of $MnO_2$/CNT (a) and $Co_3O_4$/$Mn_3O_4$ catalysts (b); Figure S9: XPS spectra of $Co^{3+}$/$Co^{2+}$ on the surface of $Co_3O_4$/$Mn_3O_4$ nano-sheet catalyst with different cobalt-manganese ratios; Figure S10: XPS spectra of $Mn^{3+}$/$Mn^{2+}$ on the surface of $Co_3O_4$/$Mn_3O_4$ nano-sheet catalyst with different cobalt-manganese ratios; Figure S11: XPS spectra of $Co_3O_4$ and Co0.9/Mn0.5 nano-catalysts; Table S1: Pore parameters of the materials; Table S2: Crystal face index, full width at half maximum, and grain size of CNT and $Co_3O_4$ nano-sheet catalysts prepared at different annealing times at 1 min, 6 min, 30 min, and 60 min [50].

**Author Contributions:** Conceptualization: X.W., J.Q. and H.W.; methodology, validation, formal analysis and investigation: X.W., K.Y., Q.Z. and J.L.; resources: H.W.; data curation: H.W.; writing— original draft preparation: X.W., K.Y., Q.Z. and J.L.; writing—review and editing: J.Q. and H.W.; visualization: H.W.; supervision: H.W.; project administration: H.W.; funding acquisition: H.W.; All authors have read and agreed to the published version of the manuscript.

**Funding:** This work was supported by the Natural Science Foundation of China under Grant (No. 21604007, H.W.).

**Data Availability Statement:** Not applicable.

**Conflicts of Interest:** The authors declare no conflict of interest.

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
