# Peer review of "Lattice Distortion in Co3O4/Mn3O4-Guided Synthesis via Carbon Nanotubes for Efficient Lean Methane Combustion"

_catalysts, doi:10.3390/catal13071112_

Round 1

Reviewer 1 Report

The authors present the catalyst preparation of Co3O4 nanosheets and their lattice-distorted Co3O4/Mn3O4 composite oxide using 1D CNT as a structure guide. The materials were used as catalysts for the oxidation of methane.

In general, the results related to the promotional effect of Mn doping on the activity are not really new, since some articles in the literature already report similar conclusions. In change, the choice of the template used in this work is novel. Also, some conclusions are only partially supported by the results; specifically, the reaction mechanism.

The XPS discussion can be improved. The authors do not give the elemental composition for some catalysts and this technique is a strictly surface analysis method.

The experimental section does not specify the atmosphere or the methodology in which the heat treatment was carried out. Was the sample heated from room temperature to the temperature of 400°C? In that section, it is said that it was kept at 400°C for 3 h, but in other sections, results with shorter heat treatments are shown. How were these treatments performed?

- Figure 1 shows HRTEM images of the Co3O4 nano-sheet. The time of heat treatment is not specified.

- The information included in Figures 2 a and 2 b is repeated in the paper. It is suggested to remove them.

- Line 110. “when cobalt nitrate 109 is oxidized to Co3O4”... Cobalt nitrate becomes Co3O4, it doesn't just oxidize. Co(NO3)2 decomposes and part of the cobalt is oxidized. The Co2+/Co3+ ratio varies with the time and temperature of the heat treatment.

-From the XRD, it is suggested to show how the crystallinity of the Co3O4 spinel changes with oxidation time. It would be beneficial for the paper to calculate the Co3O4 crystal size for all four samples.

-Section 2.2. It is recommended to expand the discussion of XPS. The results are not discussed in depth. Along with the change in the surface composition of the samples, changes are observed in the C1s spectrum area. Are these changes accompanied by any changes in the oxidation states of the cobalt species? The signals of the spectrum in the area of Co2p 3/2 and Co2p1/2 must be analyzed and deconvoluted.

-In the Co3O4 nano-sheet sample calcined for 3 h: What is the percentage of remaining carbon?

-The paragraph that begins on line 234 contains the results of various techniques. This paragraph is difficult to read. It is recommended to analyze more deeply the results obtained in each technique and separate the text into more paragraphs.

-According to the results obtained in TGA in a non-isothermal regime, the precursors with manganese present mass changes after 400°C. Could the authors show the surface composition of the Mn-containing samples? (XPS results). These results could show the percentage of CNT remaining in the samples.

-Line 269 “Recent studies have shown that higher Co3+ and higher adsorbed oxygen content are more favorable for methane catalysis [33]. In our experiment, compared with Co3O4 270 nano-sheet, doping Mn will lead to increased values of Co3+/Co2+ of Co0.9/Mn0.5 nano-sheet (confirmed by XPS analysis), which will increase the mobility of reactive oxygen species, leading to higher catalytic activity. This fact may be the reason why the catalytic activity of Co0.9/Mn0.5 nano-sheet is higher than that of Co3O4 nano-sheet. “...The XPS results related to the Co3+/Co2+ ratio have not yet been shown in the work.

-It is recommended to show tables S2 and S3 in the article.

- It would probably be beneficial for work to show O2-TPD results.

- It is not clear in the MVK proposed mechanism which site was proposed for methane adsorption and how this catalytic site is regenerated.

-It is suggested to broaden the bibliographic search and cite works that show results similar to those presented in this work. What is the advantage of using CNTs as a template?

Reviewer 2 Report

Wang et al. have synthesized Co3O4/Mn3O4 over the CNTs support for lean methane combustion. The study is interesting and it can be recommended for publication after major revisions.

Comments:

1. Abstract: First four lines in the abstract are irrelevant to the theme. This portion can be removed.

2. Please add JCPDS codes for the XRD patterns.

3. Section 2.2: As shown in TG plot, CNT is stable till 600 oC. What could be the reason for Spherical morphology formation when annealing at 400oC? Explain the reason.

4. SEM images are not depicting the 2D nano-sheets. Add suitable images or do not indicate them as nano-sheets.

5. Line 290-298: Explain it with GHSVs.

6. Section 2.4.3: Figure 8 is valid only for first-order reactions. The gas-solid reactions cannot be treated as first-order kinetics due to its non-elementary nature. The activation energies can be evaluated based on the Differential reactor approach and after verification with Weisz-Prater Criterion. Figure 8 has to be redrawn for the correct depiction. Further, the activation energies reported are impossible for methane combustion. Usually, these values are in tens to hundreds kJ/mol magnitude.

7. Assign OI and OII appropriately to explain Figure 9.

8. Often, methane combustion reaction facilitates partial oxidation of methane to form syngas due to oxygen in the reactant mixture. Report CO selectivity in the study.

9. The equilibrium lines should be added to Figure 6 and Figure S4.

1. Line 262: Correct the grammar.

2. Line 270: Is it methane catalysis or methane combustion?

Round 2

Reviewer 1 Report

The manuscript was substantially improved. This new version can be accepted for publication.

Author Response

Thank you very much to the reviewers for agreeing that our revised manuscript has been accepted and published.

Reviewer 2 Report

The authors have satisfactorily responded to most of the comments. However, following comments need to be addressed for the perfection.

1. Please maintain the correct intensities of the droplines shown in Figure S3. All the intensities of the JCPDS pattern are almost equal, which is incorect. Also, it is always better to index the reference peaks.

2. I find the apparent activation energies shown in Figure 8 are still incorrect. Following are the mistakes,

- Take Mn0.5/CNT as the reference. The Y-axis difference is about 6 units, whereas, X-axis difference is about 0.6 units. That makes the slope as 10 kJ/mol (please check the reported units). Here, the slope is the ratio of apparent activation energy (Ea) to universal gas constant. Then Ea becomes ~81.34 kJ/mol. This seems a reasonable value for methane combustion.

- The Y-axis should be rate, not rate constant (k). The units are also incorrect. Molarity (M) is reported for liquid solutions, not for gas mixtures. The units of the rate can be equivalent to mol/(g catalyst - min) or micromoles/(mg catalyst - sec).

- To find the apparent activation energy, the rates have to be considered for conversions below 10%. However, authors have considered, 100% data point also in account. At this conversion, obviously, the reactor will not be in kinetic regime and the calculations for apparent activation energy are not valid.

- The entire section 2.4.3 has to be modified accordingly.

I hope authors will take significant care on reporting the kinetics.

3. Equilibrium lines are different from T90 lines. Please go through the fundamentals and add equilibrium conversion points corresponding to the operating temperatures. 
